

# Higher-form symmetry and chiral transport in real-time Abelian lattice gauge theory

**Arpit Das[1⋆], Adrien Florio[2†], Nabil Iqbal[1‡] and Napat Poovuttikul[3∘]**

**1** Centre for Particle Theory, Department of Mathematical Sciences,
Durham University, South Road, Durham DH1 3LE, UK
**2** Department of Physics, Brookhaven National Laboratory,
Upton, New York 11973-5000, USA
**3** High Energy Physics Research Unit, Department of Physics, Faculty of Science
Chulalongkorn University, Bangkok 10330, Thailand

⋆ arpit.das@durham.ac.uk , † aflorio@bnl.gov ,
‡ nabil.iqbal@durham.ac.uk , ∘ napat.po@chula.ac.th

## Abstract

We study classical lattice simulations of theories of electrodynamics coupled to charged matter at finite temperature, interpreting them using the higher-form symmetry formulation of magnetohydrodynamics (MHD). We compute transport coefficients using classical Kubo formulas on the lattice and show that the properties of the simulated plasma are in complete agreement with the predictions from effective field theories. In particular, the higher-form formulation allows us to understand from hydrodynamic considerations the relaxation rate of axial charge in the chiral plasma observed in previous simulations. A key point is that the resistivity of the plasma – defined in terms of Kubo formulas for the electric field in the 1-form formulation of MHD – remains a well-defined and predictive quantity at strong electromagnetic coupling. However, the Kubo formulas used to define the conventional conductivity vanish at low frequencies due to electrodynamic fluctuations, and thus the concept of the conductivity of a gauged electric current must be interpreted with care.

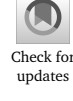

# 1  Introduction

Essentially any interacting system at finite temperature is described at long distances and times by *hydrodynamics*, i.e. by a classical theory describing the evolution of coarse-grained degrees of freedom (e.g. the fluid velocity or charge density) close to thermal equilibirum. As it is typically only conserved charges that evolve slowly on such time scales, any discussion of hydrodynamics invariably involves a careful understanding of the symmetry principles that underly these conserved charges. In this work we will study the hydrodynamic limit of lattice $U(1)$ gauge theory close to finite temperature.

In fact, recent developments in quantum field theory have resulted in new generalizations of the very idea of symmmetry itself. One such generalization is a *higher-form symmetry* [1]. Just as a conventional symmetry is generally associated – through Noether's theorem – with the conservation of a density of particles, a higher-form symmetry can be understood as the symmetry principle associated with the conservation of a density of *extended objects*, such as strings.[1]

Many very familiar theories exhibit such a global symmetry. One such system is conventional electrodynamics in four dimensions, coupled to electrically charged matter; the conserved strings in question are simply magnetic field lines, and the higher-form symmetry can be understood as the dynamical principle that ensures that magnetic field lines do not end.

This provides a novel and practical perspective on the phases of electrodynamics. In particular, in recent work this symmetry principle was used to provide a new formulation of relativistic magnetohydrodynamics (MHD) in terms of the realization of a higher form symmetry in thermal equilibrium [9]. We will briefly review the key ideas below, but we note here that the key advantage of this formalism is that it allows a formulation of MHD that is constrained only by principles of symmetry and effective field theory (EFT), and thus works perfectly well even

---

[1]See [2–8] for recent reviews on this rapidly expanding field.

when the underlying microscopic electrodynamic theory is strongly coupled. Importantly, in this formulation it is the *resistivity* of the plasma – and not the electrical *conductivity* – that is a natural transport coefficient in the effective theory.

In this work we study classical lattice simulations of finite-temperature scalar quantum electrodynamics as a representative of the MHD universality class. We measure the resistivity of this system and show that it usefully characterizes the long-range physics. We also show that within our simulations, there is no meaningful ways of defining the electric conductivity: in particular we explicitly show that the standard Kubo formula relating the conductivity to the two point function of the electric current always gives zero, being suppressed by the fluctuations of dynamical electromagnetic fields. We discuss this subtle point further and explain how a well-defined conductivity, inverse to the resistivity, can arise within the EFT framework at weak electromagnetic coupling and for long-lived electric field.

We conclude by showing that these considerations have the potential of being phenomenologically relevant. In particular, we focus on a theory where axial charge is not conserved due to an Adler-Bell-Jackiw (ABJ) anomaly [10, 11]. The effective description of a dynamically electromagnetic plasma with such an anomaly is usually called *chiral magnetohydrodynamics* (see e.g. [12–14]) Recent developments in generalized symmetry have led to advances in formalizing our understanding of anomalous transport from the point of view of EFTs and hydrodynamics [15, 16]. We show that this progress can be used to explain curious features of simulations of the chiral decay rate [17, 18] from MHD, using the resistivity determined in this work.

The manuscipt is organised as follows. In Section 2, we provide a brief background information on the symmetry-guided EFT construction of a theory with ordinary (0-form) and 1-form symmetry as well as a short introduction to the classical lattice simulation. In Section 3 we analyse relations, or lack thereof, between the conductivity $\sigma$ and resistivity $\rho$ that appear in effective description of a theory with 0-form and 1-form symmetry respectively. More specifically, Section 3.1 focus on the lack of meaningful relation among them as illustrated in the lattice simulation in a strong coupling regime while Section 3.2 illustrated on how the usual relation among them emerges when the electric field is long-lived. In Section 4, we employ the above insight to analyse the chiral decay rate of a lattice simulation with addition chiral $U(1)$ that suffers from a ABJ anomaly. This is also where we show that such decay rate is controlled by the resistivity $\rho$ and not $\sigma$ in a regime away from the weak electromagnetic coupling.

## 2 Background

### 2.1 Higher-form symmetries and hydrodynamics

In this section we briefly review the formulation of relativistic magnetohydrodynamics from the point of view of higher-form symmetry [9]. We take a somewhat leisurely route to highlight the parallels with conventional hydrodynamics; the reader who is in a hurry may skip quickly to Section 3.

#### 2.1.1 Brief review of ordinary hydrodynamics

We begin by noting that modern hydrodynamics can be usefully framed as an effective field theory. A central role in hydrodynamics is played by the conserved currents, as these evolve slowly compared to microcopic scales. The structure of the hydrodynamic theory is generally completely dictated by the global symmetries and their realization at finite temperature (see e.g. [19] for a review). To orient ourselves, we briefly recall how this works for a the familiar example of a relativistic system – e.g. an interacting complex scalar field – with a conventional

$U(1)$ global symmetry. A microscopic Lagrangian for the system might take the form:

$$S = \int d^4x \left[ -(\partial^\mu \phi)^*(\partial_\mu \phi) + V(|\phi|) \right].\tag{1}$$

The hydrodynamic regime can be thought of as a late time limit where the dynamical evolution are governed entirely by the conserved charges or densities. The hydrodynamic description can be obtained by expressing these densities in a gradient expansion of the temperature and the canonical conjugate of the conserved current. Using this scheme, we can write the conserved current $j^\mu(x)$ as

$$j^t = \chi \mu + \cdots, \qquad j^i = -\sigma \left( \partial_i \mu - E_i^{\text{ext}} \right) + \cdots.\tag{2}$$

In this expression we have neglected stress energy fluctutations, thus freezing the temperature $T$. $\mu(t, \vec{x})$ is the local chemical potential, and is the basic degree of freedom. $\chi$ – the charge susceptibility – is a thermodynamic quantity. The expression for $j^i$ in terms of gradients of the chemical potential is sometimes called Fick's law. $E_i^{\text{ext}}$ is an external applied electric field, and $\sigma$ is a transport coefficient called the *conductivity*, which will play two important roles in what follows. First, it determines the diffusion constant of the system: imposing current conservation $\partial_\mu j^\mu = 0$ and setting $E_i = 0$, we find the following dispersion relation for the diffusion of charge:

$$\mu(t,x) \sim \mu_0 e^{-i\omega t + ikx}, \qquad \omega = -iDk^2, \qquad D = \frac{\sigma}{\chi}.\tag{3}$$

It also determines the amount of current flow in response to an applied electric field. This leads to the Kubo formula which allows one to compute $\sigma$ in terms of a real-time current correlation function:

$$\sigma = \lim_{\omega \to 0} \left( \frac{1}{-i\omega} G_R^{j^x, j^x}(\omega, \vec{k} = 0) \right).\tag{4}$$

It is familiar yet non-trivial statement about hydrodynamics that the quantity obtained from the Kubo formula in (4) determines real-time dynamics as in (3).

### 2.1.2 Relativistic magnetohydrodynamics and higher-form symmetry

We now turn to the question of interest in this work, the description of relativistic *magneto*hydrodynamics in terms of symmetry principles. For an illustrative microscopic description consider the quantum field theory of Maxwell electrodynamics in four dimensions, coupled to electrically charged matter, as described e.g. by the following action:

$$S = \int d^4x \left[ -(D^\mu \phi)^*(D_\mu \phi) + V(|\phi|) - \frac{1}{4e^2} F^{\mu\nu} F_{\mu\nu} \right],\tag{5}$$

with $D_\mu \phi = \partial_\mu \phi - iA_\mu \phi$ and $F_{\mu\nu} = \partial_\mu A_\nu - \partial_\nu A_\mu$.

Now we place this theory at finite temperature. The degrees of freedom of a thermally excited plasma are electrically charged particles, interacting via electric and magnetic fields. We would like to understand the universal hydrodynamic theory describing the infrared finite-temperature physics. This framework is usually called relativistic magnetohydodynamics (see e.g. [20] for a review). It is traditionally constructed by considering Maxwell's equations coupled to a charge current that is assumed to be in thermal equilibrum as in the previous section, i.e. we write an equation of motion of the form

$$\frac{1}{e^2} \partial_\mu F^{\mu\nu} = j^\nu_{\text{dyn}},\tag{6}$$

where $j^{\mu}_{\text{dyn}}$ is a conventional dynamical electric charge current determined from an expression such as (2).

Note however that such a construction relies on knowledge of the microscopic equations of motion, and implicitly requires the existence of a separation between the electromagnetic degrees of freedom $A_{\mu}$ and the thermalized $\phi$ degrees of freedom. Such a separation may be well-justified if the electromagnetic coupling $e$ is weak; however in this work we would like to study systems where $e$ is generally $\mathcal{O}(1)$, and is not parametrically small in any sense.

More generally, it would be conceptually satisfying to have a construction of MHD that relies only on *global* symmetries and does not require any access to microscopic degrees of freedom such as $A_{\mu}$. Such a formulation is made possible by the understanding of higher-form global symmetries [1]. Indeed, as mentioned above, the global symmetry of Maxwell electrodynamics is a higher-form symmetry associated with the conservation of magnetic flux lines. This global symmetry results in a conserved current $J^{\mu\nu}$:

$$\partial_{\mu}J^{\mu\nu} = 0, \qquad J^{\mu\nu} = \frac{1}{2}\epsilon^{\mu\nu\alpha\beta}F_{\alpha\beta}. \tag{7}$$

In the language of [1] this is a *1-form* symmetry. (Conventional symmetries associated with conserved particle numbers as in (2) are 0-form symmetries.) This 1-form symmetry is the true global symmetry of electromagnetism, and is a useful starting point for an understanding of the phases of electrodynamics.[2]

In particular, it was shown in [9][3] that indeed one can reformulate MHD using this higher-form symmetry – i.e. magnetic flux conservation – as the organizing principle, resulting in a framework constrained only by thermodynamic consistency and the global symmetries. Here we present only the results of the construction, directing readers to [9] for a detailed discussion. The basic idea is to treat $J^{\mu\nu}$ on the same footing as the ordinary one-index current $j^{\mu}$ discussed in the previous section. For example, it is useful to consider coupling an external 2-form source $b_{\mu\nu}$ to (5) as

$$S \to S + \int d^4x \, b_{\mu\nu}J^{\mu\nu}. \tag{8}$$

If we consider fluctuations about the thermal state with no background magnetic field as in [28], then we can expand the magnetic flux current in constitutive relations in a higher-form analogue of (2):

$$J^{ti} = \Xi\mu^i, \qquad J^{ij} = -\rho\left(\partial_i\mu_j - \partial_j\mu_i + (db)_{0ij}\right). \tag{9}$$

Here we have worked only to linear order in the magnetic field, and have ignored the stress-energy tensor; the full construction can be found in [9].

The notation here has been picked to highlight the parallel with conventional hydrodynamics in (2), and we now unpack it. First, from (7) we see that in terms of the conventional electric and magnetic fields, we have

$$J^{ti} = B^i, \qquad J^{ij} = \epsilon^{ijk}E^k. \tag{10}$$

Here $\mu_i$ is a vector-valued "chemical potential" which can be thought of as the thermodynamic variable conjugate to magnetic flux.[4] $\Xi$ is a thermodynamic parameter that relates the conserved density $B^i$ to its chemical potential: in conventional language it is the magnetic

---

[2]For example, the regular massless 4d photon can be understood as a Goldstone mode for the spontaneous breaking of this 1-form symmetry [1, 21, 22].

[3]See earlier work formulating magnetohydrodynamics in terms of strings in [23], as well as some further developments in [24–27].

[4]In fact, in elementary electrodynamics $\mu^i$ is often called $\mathbf{H}^i$, i.e. the object whose curl is given by the free charge current. This is further explained in Appendix B. Here we choose to use the notation $\mu^i$ here to highlight the analogy with a conventional chemical potential.

permeability $\mu$. As pointed out in [9] that the field strength $db$ of an applied source $b$ in (8) can be understood as an applied *external* electric charge current

$$j^{\mu}_{\text{ext}} = \epsilon^{\mu\alpha\beta\gamma}\partial_{\alpha}b_{\beta\gamma}\,. \tag{11}$$

This applied source is often called the "free charge current" in elementary electrodynamics.

Finally, $\rho$ is a transport coefficient – it is precisely the *resistivity*. Note that it plays two distinct roles. First, $\rho$ determines the response of the electric field to an applied external current density as in (11). Indeed, it can be obtained from the following Kubo formula:

$$\rho = \lim_{\omega\to 0}\left(\frac{1}{-i\omega}G_R^{J^{xy},J^{xy}}(\omega,\vec{k}=0)\right) \tag{12}$$

(This is a correlation function for the electric field, as we have $J^{xy} = E^z$). Also, if we consider the equation of motion $\partial_{\mu}J^{\mu\nu} = 0$, then (setting the source $db = 0$) we find the following diffusive dispersion relation for the magnetic field

$$B_z(t,x) \sim B_0 e^{-i\omega t + ikx}\,, \qquad \omega = -iDk^2\,, \qquad D = \frac{\rho}{\Xi}\,. \tag{13}$$

This is the familiar expression for magnetic diffusion in a plasma. It is a non-trivial statement about hydrodynamics that the quantity obtained from the Kubo formula in (12) determines real-time dynamics as in (13).

We stress that in this formalism it is the *resistivity* which is the correct transport coefficient to consider in a hydrodynamic theory involving dynamical electromagnetism. Note also that the current $j^i$ associated with the $U(1)$ phase rotations of $\phi$ is no longer associated with a global symmetry, and thus does not obviously play a role in this discussion.

## 2.2 Classical lattice simulations

As described above, MHD ought to emerge from the sole existence of local thermal equilibrium together with magnetic flux conservation, and a simple example of such a theory is given by quantum electrodynamics coupled to some matter sector close to thermal equilibrium. Of course, studying the full quantum dynamics of such a system directly from its microscopic description is not tractable to this date. Fortunately, the corresponding classical theory, regulated on a lattice, can also be in local thermal equilibrium and has magnetic flux conservation built-in. As a result, its IR dynamics is expected to be described by MHD. It can crucially be directly studied non-perturbatively through the use of classical lattice simulations.

Concretely, we consider an interacting classical theory of a complex scalar field $\phi$ coupled to an Abelian gauge field $A_{\mu}$ with the continuum action (5). The universal dynamics that we will discuss does not depend on the precise form of the potential, at least as the field is massive and so that we stay outside the Higgs phase. For the rest of this work, we use

$$V(\phi) = m^2|\phi|^2 + \lambda|\phi|^4\,, \tag{14}$$

with real $m$ and $\lambda$.

As is well known, classical field theory in thermal equilibrium is UV divergent. In the real world, this "UV catastrophe" is regulated by (and gave birth to) quantum field theory. An alternative way of regulating these divergences is to discretize space and introduce a UV cutoff in the form of the lattice spacing $a$ (see appendix D for more details). These theories are different in the UV but have the same global symmetries and are described by the same effective theory of MHD at long distances, and are thus in the same dynamic universality class. In particular, this means that the long distance physics of both theories will be the same at the qualitative level. For instance, they both exhibit the same kind of transport phenomena,

magnetic flux diffusion, etc. which are described by the same Kubo formulas. The differences between the two theories manifest themselves as different matching coefficients, i.e. *a priori* different numerical values for the transport coefficients, which are determined in a complicated manner by the UV definition of the theory and the couplings in the potential.

To take advantage of this fact, we consider the following lattice Hamiltonian

$$H = \sum_{n \in \Lambda} \left[ |\pi_n|^2 + (D^i \phi_n)^* D_i \phi_n + V(\phi_n) + \frac{1}{2e^2} \left( \vec{E}_n^2 + \vec{B}_n^2 \right) \right]. \tag{15}$$

We denote our lattice by $\Lambda$ and make it consist of $N^3$ points. The continuous field $\phi(x)$ is replaced by a discrete version $\phi_n$. The same goes for its conjugate momentum $\pi(x)$, which is discretized to $\pi_n$. We introduced the notation $(B^i)_n = \epsilon^{ijk} \Delta_j^+ (A_k)_n$ to denote the discrete equivalent of the magnetic field and $\Delta_i^+ f_n = \frac{1}{a} \left( f_{n+\hat{i}} - f_n \right)$ is a finite difference version of the continuum derivative in the $i^{th}$ direction, characterized by the unit vector $\hat{i}$. Similarly, $\vec{E}_n = (E_x, E_y, E_z)_n$ is the electric field, canonical conjugate to $\vec{A}_n$. The discrete covariant derivatives are realized thanks to the introduction of discrete parallel transporters ("links") $(U_i)_n = e^{-iae(A_i)_n}$, $D_i \phi_n = \frac{1}{a} \left( (U_i)_n \phi_{n+\hat{i}} - \phi_n \right)$. We use periodic boundary conditions for the fields $\phi_n, \pi_n$ and $E_n$. In the absence of external magnetic field, $A_n$ also has periodic boundary conditions. In the cases where we consider a background magnetic field – which is only in Section 4.2 – we implement it through twisted boundary conditions for the $A_n$ fields. We refer the interested reader to Appendix A of [29] for technical details. Note that by writing down this Hamiltonian we decided to work in temporal gauge $A_0 = 0$. In particular, it represents a constrained Hamiltonian system and needs to be supplemented by Gauss law

$$\Delta_i^+ E_n^i = 2e^2 \text{Im}(\pi \phi^*)_n, \tag{16}$$

which selects the gauge invariant subspace in field space. Note also that, crucially, this discretization automatically imposes Bianchi's identity. this discretization automatically imposes Bianchi's identity.

Classical thermal equilibrium in this system at temperature $T$ is described by the statistical partition function

$$Z[\beta] = \int \prod_{n\Lambda} d\phi_n d\pi_n d\vec{E}_n d\vec{A}_n e^{-H/T}, \tag{17}$$

$$\langle O \rangle_T = \int \prod_{n\Lambda} d\phi_n d\pi_n d\vec{E}_n d\vec{A}_n O e^{-H/T}, \tag{18}$$

with $O$ some operator and we denote by $\langle O \rangle_T$ its thermal average. In practice, all thermal averages of interest are computed by sampling field configuration from the Boltzmann distribution $e^{-H/T}$ using a standard Metropolis algorithm.

Our motivation to study this classical system is that unequal time correlation functions can directly be computed. To do so, we evolve our sampled field configurations along classical trajectories specified by the Hamiltonian dynamics

$$\partial_t \pi_n = -\frac{\partial H}{\partial \phi_n}, \qquad \partial_t \phi_n = \pi_n, \tag{19}$$

$$\partial_t \vec{E}_n = -\frac{\partial H}{\partial \vec{A}_n}, \qquad \partial_t \vec{A}_n = \vec{E}_n. \tag{20}$$

Note that here we are solving the pure classical theory. Only the IR modes have a chance of being in some thermal equilibrium state. An alternative approach would be to use an effective

Langevin theory with hard modes integrated out as stochastic noise. These trajectories give us access to classical-statistical correlators of the type $G_{cl}^{OO}(t,\vec{n}) \equiv \left\langle O(t,\vec{n})O(0,\vec{0})\right\rangle_T$ for any given operator $O$. In particular, they give us access to the classical counterparts of the Kubo formulas (12) and (4):

$$\rho = \frac{1}{3}\sum_{i=x,y,z}\int_0^\infty \mathrm{d}t \sum_{\vec{n}\in\Lambda} G_{cl}^{E_iE_i}(t,\vec{n}) \equiv \int_0^\infty \mathrm{d}t\, G_{cl}^{EE}(t)\,, \tag{21}$$

$$\sigma = \frac{1}{3}\sum_{i=x,y,z}\int_0^\infty \mathrm{d}t \sum_{\vec{n}\in\Lambda} G_{cl}^{j_i^{\mathrm{dyn}}j_i^{\mathrm{dyn}}}(t,\vec{n}) \equiv \int_0^\infty \mathrm{d}t\, G_{cl}^{j^{\mathrm{dyn}}j^{\mathrm{dyn}}}(t)\,. \tag{22}$$

Where we introduced the shorthand notations $G_{cl}^{EE}(t) = \frac{1}{3}\sum_{i=x,y,z}\sum_{\vec{n}\in\Lambda} G_{cl}^{E_iE_i}(t,\vec{n})$ for the isotropized version of the electric field two-point function at zero spatial momenta, and similarly for the electric current. $G_{cl}^{OO}(t,\vec{n})$ is the classical limit of the statistical propagator, which is related to the retarded one near equilibrium by the KMS relation, whose expression in the classical limit reads

$$G_{cl}^{OO}(\omega,\vec{n}) \approx -\frac{2T}{\omega}\mathrm{Im}\left(G_R^{OO}\right)\,. \tag{23}$$

More details on this correspondence are given in App. A.

In practice, and for the rest of this work, we work in units where the lattice spacing $a = 1$, and we set the temperature to $T = 1/a = 1$, i.e. our temperature is at the lattice scale. Let us briefly explain how to restore units to the dimensionless numerical results presented. Consider an observable $\mathcal{O}$ with mass dimension $\Delta$. On general grounds its functional dependence on all parameters will be given by

$$\mathcal{O} = T^\Delta f(Ta, ma, \cdots)\,, \tag{24}$$

where $f$ is a dimensionless function of all physical quantities measured in units of the lattice scale. Our results should be interpreted as determining the dimensionless function $f$ at a particular value of its arguments; the appropriate power of $T$ can be restored by dimensional analysis if required – e.g. if we restore units to the bottom panel of Figure 1 it would be interpreted as a plot of $\rho T$ against $tT$ – but as we are not in a continuum limit we stress that the dependence on the UV cutoff $a$ appearing in the scaling function $f$ can never be removed.

We also wish to fix the scalar mass $m$ and the coupling $\lambda$. For our problem, the only important consideration is that the parameters land the model in its unbroken phase. In order to ease comparison with Ref. [18], we adopt the same choice, namely[5] $m^2 = e^2T^2/4$ and $\lambda = \frac{e^2}{2}$. This choice is motivated by phenomenological considerations and further discussed in Ref. [18].

While we defer the details of equations (19)-(20) and a recap of our numerical schemes in App. C, we want to emphasize a feature of our discretization. First, by solving for the real-valued gauge fields $A_i$ (and *not* the parallel transporters or links), we study a non-compact $U(1)$ gauge theory. This means that there are no dynamical magnetic monopoles in the model and the continuous 1-form symmetry associated with the conservation of magnetic flux is preserved on the lattice, as manifested in Bianchi's identity. Indeed, one has, as in the continuum,

$$\sum_i \Delta_i^+ B_i = \sum_{ijk}\epsilon_{ijk}\Delta_i^+\Delta_j^+ A_k = 0\,, \tag{25}$$

$$\partial_t A_k = E_k \implies \partial_t B_i - \epsilon_{ijk}\Delta_j^+ E_k = 0\,. \tag{26}$$

These relations are together equivalent to the local conservation of the 2-form current $\partial_\mu J^{\mu\nu} = 0$ as in (7). This conservation law leads to the diffusion of the associated charge,

---

[5]We also use the improved lattice mass of [30], see [18] for more discussions.

namely magnetic flux, as we will verify explicitly in Section 3.1.3. Note however that with periodic boundary conditions the total magnetic flux threading the system is zero; hydrodynamics describes the *local* diffusion of magnetic flux.

# 3 Of resistivity and conductivity

To recap, when given a microscopic description of an electromagnetic plasma (5), it would appear that two points of view coexist to describe the long-distance dynamics. The "conventional" approach to MHD is based on electric charge conservation. It studies the dynamics of the electric charge current – which is subsequently gauged – and thus introduces the conductivity $\sigma$ as a transport coefficient, telling us how electric charges move in response to an electric field. It couples the electric charge to matter by assuming Ohm's law $\vec{j}_{\text{dyn}} = \sigma \vec{E}$ and builds dynamics around this point of view, explicitly imposing Maxwell's equations with an electromagnetic coupling $e$.

Another approach is based on the conservation of magnetic flux, and is thus directly connected to the global 1-form symmetry of dynamical electromagnetism. It directly describes the dynamics of electric and magnetic fields, and thus directly introduces resistivity $\rho$ as a transport coefficient as in (12). This describes how dynamical electric fields move in response to an external charge current. Importantly, the equations of motion for MHD close by themselves with no choice needed for dynamics of $j^i$ or explicit mention of Maxwell's equations.

As argued above, when the electromagnetic coupling $e$ is weak, these points of view are equivalent and should agree. Indeed on elementary grounds one expects a relationship of the form $\rho = \sigma^{-1}$. The precise nature of this agreement is however somewhat subtle: the definition we gave for $\sigma$ was in terms of a low-frequency limit in (4). However the low frequency limit clearly does not commute with the weak coupling $e \to 0$ limit in a theory with a hydrodynamic description.

Here we will show from direct simulations that when the electromagnetic coupling $e$ is strong – as it is in our lattice simulations – the "conventional" point of view is not practical anymore. In particular, a useful non-perturbative notion of conductivity appears to be lost, in a sense that we will make precise. On the other hand, the organization around magnetic flux conservation and in terms of resistivity $\rho$ still provides practical predictions.

In more detail, we study the classical equilibrium (15) described in the previous section and focus on its long-range dynamics. In this section we will simultaneously describe numerical results in parallel with their theoretical explanation.

## 3.1 $\rho$ and $\sigma$ from lattice simulation

We begin from the point of view of magnetic flux conservation and compute the resistivity $\rho$ from the Kubo formula (12). To this end, we need a reliable estimation of the zero momentum electric correlator. We show the results for $e^2 = 1, N = 200$ in the upper panel of Fig. 1. The correlator is obtained from an average of 500 simulations see App. C for more information about the simulation parameters. By the Kubo formula (21), the resistivity is the integral of this correlator. In practice, we define the quantity

$$\rho_t = \int_0^\infty \mathrm{d}t' G_{cl}^{EE}(t').  \tag{27}$$

As shown in the bottom panel of Fig. 1, it saturates to a constant value. Our final estimate of $\rho$ is obtained by averaging $\rho_t$ at late time $t > 200$.

We can already note that $\rho|_{e^2=1} = 1.06 \pm 0.03$. has a finite value that can be accurately determined. We also see oscillations in the propagators. They are of a non-universal origin.

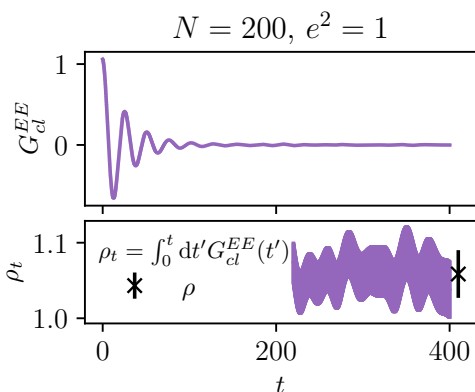

Figure 1: Electric correlator (upper panel) and extraction of resistivity $\rho$ (lower panel). The integral saturates, giving rise to a finite value for $\rho_{e^2=1} = 1.06 \pm 0.03$.

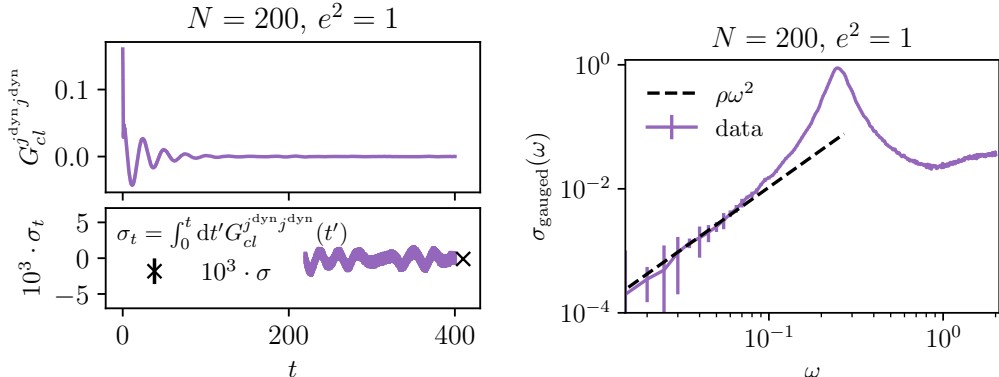

Figure 2: **Left:** Electric current correlator (upper panel) and extraction of $\sigma^{\text{gauged}}$ (lower panel). The correlator integrates to zero within statistical uncertainties, of order of a part in $10^3$ (note the rescaled axis). **Right:** Frequency dependent gauged conductivity (Fourier transform of left hand side). As argued in the main text, Maxwell's equations predicts that $\sigma^{\text{gauged}}$ goes to zero like $\rho\omega^2$. This is what is shown as a dashed line on the picture, with $\rho|_{e^2=1}$ determined above.

They are plasmon-like oscillations generated by our classical lattice equivalent of hard thermal loops. We discuss this further in Appendix D.

### 3.1.1 On extracting the conductivity when the gauge field is dynamical

In our system, we have access to the microscopic electric charge current $\vec{j}_{\text{dyn}}$. It thus allows us to directly compute an associated electric conductivity. We proceed in the same way as with the resistivity: we compute the current-current correlator $G_{cl}^{j^{\text{dyn}}j^{\text{dyn}}}$ and define $\sigma_t^{\text{gauged}} = \int_0^\infty dt' G_{cl}^{j^{\text{dyn}}j^{\text{dyn}}}(t')$. The superscript indicates that this quantity is calculated in the gauged theory from the Kubo formula (4). We show the results in the left panel of Fig. 2. We find that the conductivity, defined from the Kubo formula, vanishes in this system $\sigma^{\text{gauged}}|_{e^2=1} = -000010 \pm 0.00014$.

The vanishing of the conductivity might seem surprising but can easily be explained. It is a direct consequence of Maxwell's equations. The current reads

$$e^2 j_i^{\text{dyn}} = \partial_t E_i + (\nabla \times B)_i \,. \tag{28}$$

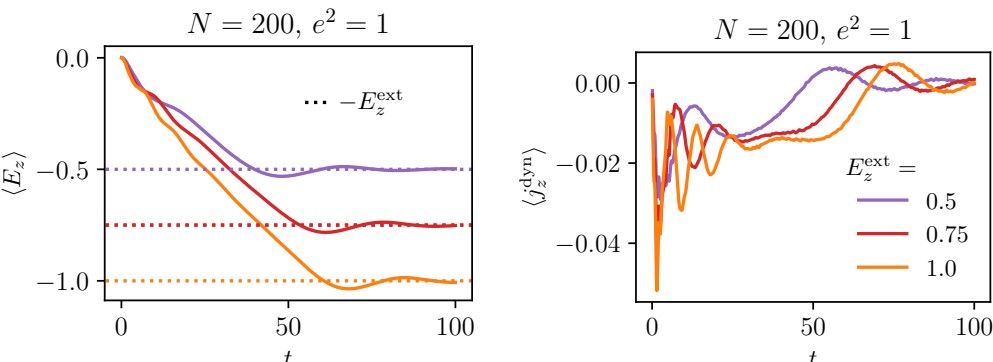

Figure 3: **Left:** Mean electric field along the $z$ direction subject to an external electric field quench in the $z$ direction. The quench we implement simply corresponds to a change in the initial conditions of the electric fields. As a result of electric screening, it quickly relaxed to the external field value. **Right:** Mean dynamical electric current subject to an external electric field quench in the $z$ direction. As the electric current is sourced by the time derivative of the electric field which is constant at late time, it is not possible to source a non-zero steady current in this way. We observe the emergence of an approximately steady current at intermediate times. For less strongly interacting systems, this intermediate regime is the one giving rise to well-defined electric conductivity.

This implies the following relation between the electric and electric current correlator at zero spatial momentum

$$G_{cl}^{j^{\mathrm{dyn}} j^{\mathrm{dyn}}}(\omega, k = 0) = -\frac{\omega^2}{e^4} G_{cl}^{EE}(\omega, k = 0). \tag{29}$$

The left-hand side determines the resistivity through the Kubo formula (12). We see that as long as the resistivity is non-zero and finite, the extra factors of $\omega$ on the right-hand side mean that the Kubo formula for conductivity necessarily vanishes in the presence of dynamical electromagnetic fields.

Physically, this is a consequence of electric screening. The conductivity measures the linear response to an applied external electric field. If electromagnetism is dynamical, there is no precise meaning to the concept of an "external electric field", indeed any putative external electric field will always be screened by a dynamical one, resulting in a vanishing $j^{\mathrm{dyn}}$ at late times. This is illustrated in Fig. 3, where we attempt to add an external electric field. There is no completely canonical way to do this in a theory of dynamical electromagnetism, but in Appendix C we demonstrate a physically reasonable scheme.

Note that the dynamical electric field asymptotes to a constant value which precisely cancels the external one. Recalling Maxwell's equations, the time derivative of the dynamical electric field sources the electromagnetic current. An electric current is then produced while the dynamical electric field readjusts to screen the external one but vanishes eventually, in the steady state.

Given these results, one may be confused about the meaning of the conductivity that is often computed from the current-current two point function in a theory of dynamical electromagnetism, e.g. as in [31, 32]. It seems that giving a precise meaning to $\sigma$ involves an order of limits. The Kubo formula for conductivity assumes that the probe electric field is non-dynamical, and that the electric field is decoupled. This notion of decoupling acquires meaning in the dynamical theory only if the electric field evolves on a timescale $\tau_E$ that is sufficiently long that the electric field can be thought of as being fixed over the timescale of

the conductivity measurement. In Section 3.2 we will estimate $\tau_E$ and show that it depends on the electromagnetic coupling; the required hierarchy of scales does not exist if $e$ is large.

Operationally, this can be thought of in light of linear response and Fig. 3. A transient current appears for the time it takes to screen the external electric field. If the electric field is sufficiently long-lived, the linear response regime can be reached. The dynamical electric field slowly varies to screen the external one. This in turns creates an approximately constant current and the conductivity could in principle be extracted from this transient current. It is also clear from Fig. 3 that this separation of scales does not happen in our current system. The external field is swiftly quenched and no current is produced.

### 3.1.2 Linear response and resistivity

Before moving on to this, we explore further the consequence of magnetic flux conservation. In particular, it is interesting to consider the meaning of resistivity from the point of view of linear response theory. As recalled in (12)-(11), $\rho$ measures the linear response of the electric field to an external charged current. Physically, from the point of view of magnetic fluxes, a current of probe charges rearranges magnetic field lines, which in turns produce some electric field by induction. This phenomenon is illustrated in Fig. 4, where we study the time evolution of our system in the presence of a constant external current $j^{\text{ext}}$ of charges along the third direction as defined in (11); see App. C for further details.

On the left, we show the mean value of the electric field along the third direction. As expected, the field quickly reaches some constant value. The dashed line is the prediction of linear response, using the resistivity from the Kubo formula (27). The agreement is impressive.

On the right-hand side, we show the average value of the dynamical electromagnetic current. To create a stable steady state, it has to cancel the applied external one, and it does. Note also that in this way one effectively recovers a version of Ohm's law, whereby

$$\langle j_i^{\text{dyn}} \rangle = -j_i^{\text{ext}} = -\frac{1}{\rho} \langle E_i \rangle \,. \tag{30}$$

In this sense, one can always define conductivity as the inverse of resistivity, computed from the Kubo formula of the electric field (21). However, as discussed above and in section 3.2, it acquires a meaning of its own as the response of charged matter to an external electric field only at weak coupling.

### 3.1.3 Magnetic diffusion

Finally, we also examine magnetic diffusion, as predicted by Eq. (13). This is illustrated in Fig. 5, where we consider the time-dependence of the magnetic field correlator at different values of spatial momentum $k$. While the $k^2$ dependence of the exponent is clear, and the qualitative behavior of the correlator is the one expected and compatible with $\rho \approx 1.06$, a quantitative analysis proves harder to conduct. The main limiting factor is that an independent extraction of the magnetic susceptibility $\Xi$ – which measures fluctuations of magnetic flux which wraps the whole system – is hard in the presence of periodic boundary conditions. While the local fluctuations of magnetic fluxes are unconstrained, the total fluxes in the box are frozen to zero as a result of the local nature of our Monte Carlo updates. While in principle it is possible to extract the susceptibility from local measurements – see for instance [33] for related discussions in the context of the QCD topological susceptibility – it is beyond the scope of this work and of no intrinsic value per se. We see that using the bare value of $\Xi = e^2 = 1$ describes the data fairly well, suggesting that this parameter is only weakly renormalized if at all.

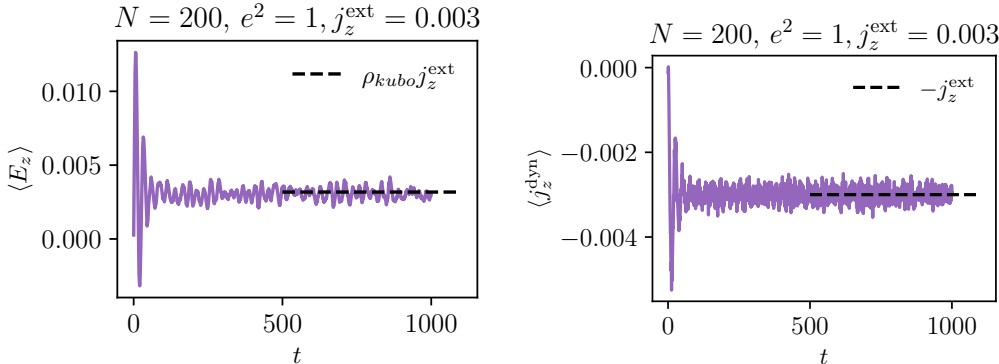

Figure 4: **Left:** Mean electric field along the $z$ direction in the presence of an external electric current along the $z$ direction. As predicted by linear response, a nonzero electric field is generated, whose magnitude is perfectly predicted by the resistivity determined above from its Kubo formula. **Right:** Mean dynamical electric current along the $z$ direction in the presence of an external electric field along the $z$ direction. A steady state is created by creating a flow of charge in the opposite direction to the external probe.

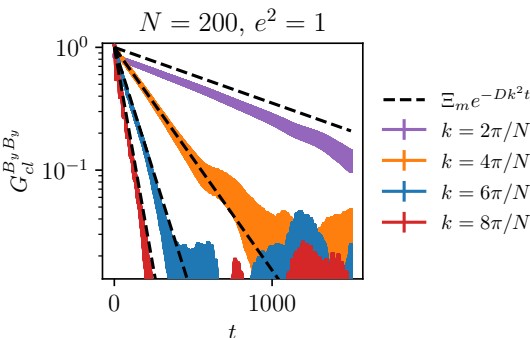

Figure 5: Transverse magnetic correlator for different momenta. The dashed line corresponds to the naive expectation $G_{cl}^{B_y} \sim e^{-Dk^2 t}$ with $D = \rho/\Xi$, $\rho$ is extracted from the left-hand side. As explained in the text, $\Xi$ is difficult to extract independently with periodic boundary conditions. It does appear that it is only weakly renormalized from the bare value $\Xi = 1$ which we use in this plot. The $k^2$ dependence is clear.

## 3.2 Electric field relaxation at weak coupling

As discussed in the previous section, a key role in interpreting the conductivity is played by the relaxation time of the electric field. Here we study this phenomenon quantitatively by introducing a generalization of the effective theory of MHD and comparing it to weak coupling.

The required formalism is a modification of magnetohydrodynamics that allows for a finite electric field lifetime, as discussed in [34] (see also [35]). Following that work, we have an expression for the slightly broken conservation of the electric field, which is a generalization of (9).

$$\partial_t J^{ti} + \partial_j J^{ji} = 0, \qquad \partial_t J^{ij} + \frac{\rho}{\tau_E}\left(\partial_i \mu_j - \partial_j \mu_i + H_{tij}\right) = -\frac{1}{\tau_E} J^{ij}. \tag{31}$$

The new parameter $\tau_E$ controls the decay rate of the electric field via $J^{ij}(t) \sim e^{-t/\tau_E}$. The universal formulation of MHD in terms of 1-form global symmetry reviewed in Section 2.1.2 corresponds to an effective description for long times $t \gg \tau_E$; in this case the term in $\partial_t J^{ij}$

may be neglected, and $J^{ij}$ has no independent dynamics, being fully determined by gradients of $\mu^i$. Note that $\tau_E$ is an independent parameter and there is no *a priori* relation between the coefficient $\rho$ and the decay rate $1/\tau_E$.

On the other hand, the equations of motion of the *conventional* formulation of plasma, in e.g. [20] where the electromagnetic field is weakly coupled to the matter sector – can be cast precisely in the form (31). In Appendix B we perform a careful matching and find the relations

$$\tau_E = \frac{1}{\sigma e^2}, \qquad \rho = \frac{1}{\sigma} \tag{32}$$

(In this expression $\sigma$ is defined in the conventional way, as the quantity apearing in Ohm's law that relates the dynamical current $j^{\mathrm{dyn}}$ to the dynamical electric field $E$; see Appendix B for details).

We see that the basic differentiating ingredient in a "conventional" formulation of the plasma is that the electric field is introduced as a degree of freedom, with its relaxation time $\tau_E$ determined by the microscopic coupling $e$. Note that hydrodynamics is valid for $t \gg \tau_E$, and that as expected this timescale diverges as we take the coupling to zero.

Within the theory defined by (31) we can compute various two-point functions. We find that the resistivity $\rho$ can be written[6] as

$$\rho = \lim_{\omega \to 0} \frac{1}{\omega} \mathrm{Im} G_R^{J^{xy} J^{xy}}(\omega, \vec{k} = 0), \qquad G_R^{J^{xy} J^{xy}}(\omega, \vec{k} = 0) = \frac{\omega \rho (i - \omega \tau_E)}{1 + \omega^2 \tau_E^2}. \tag{33}$$

It is instructive to compute the dynamical current-current correlation function for the conductivity of $j_i^{\mathrm{dyn}}$ defined from (11). Using $\partial_t n^{\mathrm{dyn}} + \partial_i j^{\mathrm{dyn}\, i} = 0$ and $n_{el} = e^{-2} \partial_i E_i$, from (31) we find that

$$G_R^{j_z^{\mathrm{dyn}} j_z^{\mathrm{dyn}}}(\omega, \vec{k} = 0) = \lim_{k_z \to 0} \frac{\omega^2}{k_z^2} G_R^{n^{\mathrm{dyn}} n^{\mathrm{dyn}}}(\omega, k_z) = \frac{1}{e^4} \frac{\rho \omega^3}{i + \omega \tau_E}. \tag{34}$$

We can use this to define the conductivity $\sigma_{\mathrm{gauged}}$ of the electric charge current in a theory of dynamical electromagnetism as

$$\sigma_{\mathrm{gauged}}(\omega) \equiv \frac{1}{\omega} \mathrm{Im} G_R^{j_z^{\mathrm{dyn}} j_z^{\mathrm{dyn}}}(\omega, \vec{k} = 0) = \frac{1}{e^4} \frac{\rho \omega^2}{1 + \omega^2 \tau_E^2}. \tag{35}$$

The vanishing of this quantity at low frequencies was previously shown in (29) and seen numerically in Figure 2. Here we have included the effects of a finite electric field lifetime $\tau_E$. Indeed if we now match $\tau_E$ to a microscopic description using (32) (and further assume that the electric field is the only longest-lived nonconserved operator), then we can extrapolate $\omega \gg 1/\tau_E$ and find the expression

$$\sigma_{\mathrm{gauged}}(\omega \gg 1/\tau_E) = \frac{1}{e^4} \frac{\rho}{\tau_E^2} = \sigma, \tag{36}$$

where in the last equality we have used (32).

Thus the effects of dynamical electrodynamics soften the correlator, but if one goes to higher frequencies, in this simple framework with one timescale one can extract a finite result for the conductivity. This regime exists only over intermediate frequency scales $\tau_E^{-1} \ll \omega \ll \Lambda$, where $\Lambda$ denotes a microscopic scale in the theory beyond which hydrodynamics is not valid. Thus the existence of the regime requires a hierarchy of scales, which is indeed expected to exist in weakly coupled electrodynamics due to (32).

---

[6]On the other hand, the parameter $1/\tau_E$ can be obtained from $G_R^{\partial_t J^{xy} \partial_t J^{xy}}(\omega, \vec{k} = 0)$ via the memory matrix formalism [34] with no prior relation to $\rho$.

It seems that the classic perturbative (in $e$) calculations of $\sigma$ in a plasma of dynamical electromagnetism (see e.g. [31, 32]) should be interpreted as extracting $\sigma$ in this regime. We are not aware of any perturbative calculation that explicitly shows the crossover to the vanishing value of $\sigma^{\text{gauged}}$. Note that through (32) the value of $\sigma$ in this regime is indeed numerically equal to $\rho^{-1}$, and thus – provided the hierarchy of frequency scales exists – this approach should correctly provide a characterization of the plasma.

We attempted to investigate this crossover in our simulations. However we found that the presence of a sharp gapped resonance, which we interpret as a plasmon – as shown in the right panel of Figure 2 – made it impossible to enter a regime where the correlation function of $j^{\text{dyn}}$ saturates to a constant as in (36). In the language above the plasmon frequency plays the role of $\Lambda$ and the required hierarchy does not exist.

## 4 Application: Chiral transport

We conclude this work with a concrete application where these considerations may have an impact: chiral transport. We start by briefly discussing chiral transport from the point of hydrodynamics before presenting our numerical results.

### 4.1 Background: Chiral magnetohydrodynamics

Consider the theory of electrodynamics coupled to massless Dirac fermions, i.e.

$$S = \int d^4x \left[ \bar{\psi} \slashed{D} \psi - \frac{1}{4e^2} F^{\mu\nu} F_{\mu\nu} \right] + S_\phi \tag{37}$$

(The inclusion of a charged scalar field in $S_\phi$ does not modify the universality class, and we include it for later convenience). This theory now has extra structure compared to (5): in particular its 0-form axial current $j_A^\mu \equiv \bar{\psi}\gamma^5\gamma^\mu\psi$ is not conserved at the quantum level because of the Adler-Bell-Jackiw anomaly:

$$\partial_\mu j_A^\mu = \kappa \epsilon^{\mu\nu\rho\sigma} J_{\mu\nu} J_{\rho\sigma} , \tag{38}$$

where $\kappa = \frac{1}{16\pi^2}$ is an anomaly coefficient. Note that we have chosen to express the non-conservation in terms of the 2-form symmetry current in (7). This highlights that the breaking of the symmetry due to this anomaly is in fact a kind of intertwining of the 0-form axial symmetry and the 1-form magnetic flux symmetry. Indeed, it was recently explained that the most precise characterization of this structure is in terms of a *non-invertible symmetry* [36, 37],[7] which lets one construct conserved axial charge operators that are deformed by the anomaly to obey a composition law which is not that of a normal $U(1)$ group.

One can now place this theory at finite temperature and ask about the long distance hydrodynamic behavior. The resulting framework is called chiral magnetohydrodynamics. Despite intense study due to its phenomenological importance – see e.g. [14, 40–42] – questions remain. Much of the literature predates the recent refined understanding of symmetry structure and is not framed in the discussion of effective field theory, leading to potential confusion about the domain of validity of the resulting theory.

We briefly review the conventional approach to this problem; one splits the theory into a Maxwell sector and a matter sector, and imposes Maxwell's equations in the form (2), assuming that the the gauged $U(1)$ current $j^{\text{dyn}}$ takes the form

$$j_i^{\text{dyn}} = \sigma E_i + 8\kappa \mu_A B_i , \tag{39}$$

---

[7]For an alternative way to write the non-invertible symmetry generator, see [38, 39]. See also [8] for a recent review on the non-invertible symmetry of this type.

where the second term in the axial chemical potential $\mu_A$ arises due to the anomaly in the ungauged theory (see e.g. [43] for a review).

As explained extensively above in the context of ordinary MHD, one may expect difficulties with this framework if the electromagnetic coupling is large and $\sigma$ is difficult to define.

For example, a basic quantity of interest is the axial charge relaxation rate. Due to the anomaly, the axial charge density $n_A$ is no longer conserved, and instead decays with a rate $\Gamma_A$. One can attempt to obtain an expression for $\Gamma_A$ from elementary hydrodynamic arguments [17], relating it to the electric conductivity (4) of the ungauged theory. See Section 4.2 for more discussion. Fasciatingly, previous real-time simulations [17, 18] appear to disagree with this formula; in particular, a reasonable estimate for the electric conductivity results in a decay rate that is off from the hydrodynamic prediction by about an order of magnitude.

We now turn to more recent work an effective field theory framework, including [15, 16]. These theories allow for the computation of various observables: In particular, [15] obtained an expression for $\Gamma_A$ in terms of the resistivity $\rho$. This leads to a universal formula for this decay rate at small magnetic field:

$$\Gamma_A = \frac{64\kappa^2}{\chi_A} B^2 \rho \,. \tag{40}$$

Importantly, here the resistivity $\rho$ is expressed in terms of the Kubo formula (12).

Though to the best of our knowledge this relation was first expressed in this form following hydrodynamic considerations in [15], the Kubo formula (40) is not a surprise. It can also be obtained from a more microscopic point of view as consequence of the fluctuation-dissipation theorem which relates the chiral decay rate to the Chern-Simons diffusion rate – see Appendix B of [18] for a derivation. The Kubo formula follows from equation (B.17) there, where the input from hydrodynamics is in the interpretation of the correlation function of the electric field in terms of the resistivity.

A holographic study in the same universality class was performed in [44], and demonstrated agreement with this relation. In this work we demonstrate that (40) is in perfect agreement with observations of the axial charge decay rate in real-time simulations, thus resolving the discrepancy noted in [17, 18].

## 4.2 Numerical results on the chiral decay rate

Direct computations of anomalous transport using semiclassical methods from a microscopic theory are possible [45–49] but computationally very challenging. Ref. [18] follows a different approach. An anomalous sector is added in an effective way to the microscopic action (5), leading to a hybrid microscopic-effective model. An additional homogeneous degree of freedom $a(t)$ is added to the theory as follows

$$S_a = S + \int \mathrm{d}x^3 \left( \frac{\chi_A}{2} \partial_t a \partial_t a + \kappa a \epsilon_{\mu\nu\rho\sigma} F^{\mu\nu} F^{\rho\sigma} \right) \,. \tag{41}$$

The time derivative of $a$ plays the role of the axial chemical potential $\partial_t a = \mu_A$. In particular, Ref. [18] takes $\kappa = \frac{1}{16\pi^2}$ and $\chi_A = \frac{T^2}{3}$ as motivated by QED.[8] We use the same parameters in the subsequent analysis. The homogenous dynamics of the axial charge is expected to be described by the anomalous MHD framework of [15].

---

[8]We use slightly different notations from Ref. [18] in order to make more direct contact to Ref. [15]. In particular, writing $\mathfrak{a}, \mathfrak{u}_5$ the "axion" and the chemical potential of [18], we have $a = \frac{\mathfrak{a}}{2\Lambda}$, $\mu_A = \frac{\mathfrak{u}_5}{2}$ and indeed $\chi_A = 4\Lambda^2 = \frac{T^2}{3}$, with $\Lambda$ the axionic coupling of [18].

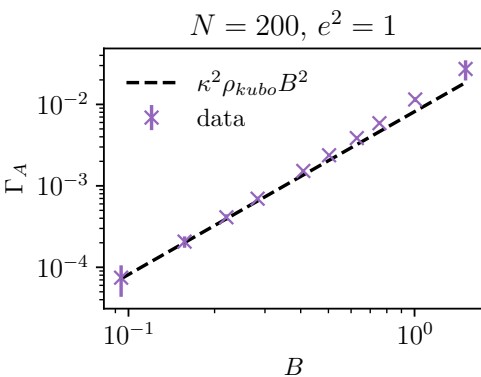

Figure 6: Chiral decay rate as a function of the external magnetic field. The linear response prediction in terms of the resistivity is shown as a dashed line; the agreement is impressive. We also show the dependence of the rate for stronger field and see a departure from the $B^2$ dependence.

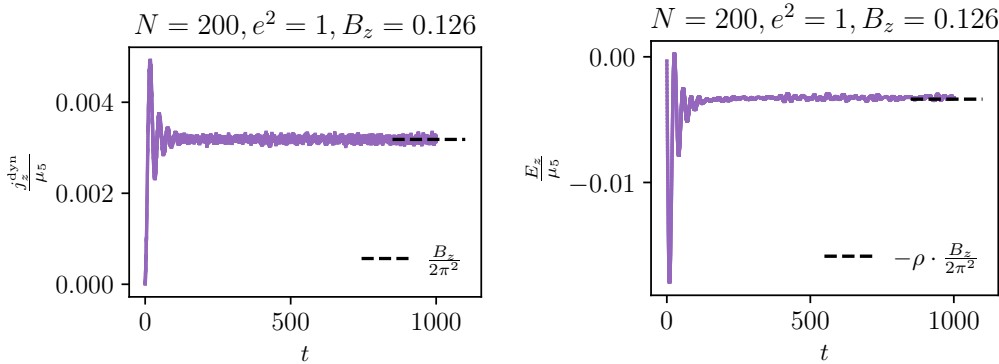

Figure 7: **Left:** Ratio of the dynamical current generated by the decay of the chiral chemical potential to the chiral chemical potential. While both $j_z^{\text{dyn}}$ and $\mu_A$ are time dependent, their ratio is constant. **Right:** Ratio of the electric field to the chiral chemical potential. We see that it also well described by linear response theory and simply related to the chiral magnetic current by the resistivity determined in the previous section.

The effective chiral chemical potential is conjugate to the chiral charge density in the full microscopic theory $n_A = \chi_A \mu_A$. Indeed, its dynamics is dictated by the following "anomaly" equation

$$\chi_A \dot{\mu}_A = \kappa F_{\mu\nu} \tilde{F}^{\mu\nu}. \tag{42}$$

The backreaction into the gauge fields equations happens through the generation of a chiral magnetic current $-\frac{1}{2\pi^2}\mu_A \vec{B}$. The scalar sector is not affected by these modifications. Its role is to simply provide a microscopic implementation of a electrically charged matter sector.

This theory allows us to directly check the Kubo formula (40). We start by computing again $\Gamma_A$ by fitting the exponential decay of the chemical potential $\mu_A$, see [18] for more information. After checking that our results are in agreement, we extent the determination of $\Gamma_A$ to larger magnetic fields. We show the results in Fig. 6. The dashed line corresponds to the prediction of the Kubo formula (40), using the resistivity computed in Fig. 1. The agreement up to moderate values of $B$ is impressive. The deviations at larger values of $B^2$ simply signal the breaking of linear response.

It also allows us to illustrate a simple physical insight coming from these considerations. The decay of the axial charge is easily explained once the presence of a chiral magnetic current is known. Equation (40) is immediate if one assumes that the electric field is related to the chiral magnetic current through linear response $\vec{E} = \rho \, j_{CME}^{\text{dyn}} = \rho \frac{1}{2\pi^2} \mu_A \vec{B}$ and inserts it in the anomaly equation (42). The elementary derivation mentioned in Section 4.1 proceeds in the same way except it assumes Ohm's law $j_{CME}^{\text{dyn}} = \sigma \vec{E}$ and predicts $\Gamma_A^\sigma = \frac{64\kappa^2}{\chi_A} \frac{1}{\sigma} B^2$. The conceptual difference between the two is whether linear response should be applied to the electric field or the current. Our results simply show that when $\sigma$ cannot be unambiguously defined, the universal approach is to consider the electric field as being sourced from the chiral magnetic current.

We confirm that this is what happens in our system on the left-hand side of Fig. 7, where we plot the ratio of the mean electric current to the chemical potential. We see that this ratio is well described by the CME prediction. More to the point, we see that it then induces a constant response electric field $\vec{E} = \rho \, j_{CME}^{\text{dyn}}$.

To conclude, we verify that these results hold for different values of the electromagnetic coupling $e$. Concisely, we compare the $e$-dependence of $\rho$ obtained through three different methods: from the $\Gamma_A$ data of [18], by fitting the linear response of $E$ to an external current and from the Kubo formula (12).

We demonstrate the results in Fig. 8. Let us start by commenting that the extraction using the Kubo formula is much more costly than the linear response extraction. It requires averages over hundred of samples to extract a signal, compared to a single sample. This explains why we generated more charges for the "quench" data. Second, the extraction from $\Gamma_A$ and the direct linear response are in alsmost perfect agreement. This further supports the above explanation; the electric field created from the chiral decay is a response to the chiral magnetic current. The agreement with the Kubo formula is also very good. The few percent discrepancy at larger charges can be taken as an assessment of our systematic errors. For instance, the linear response regime decreases at larger charge, making the extraction of $\rho^{quench}$ and $\Gamma_A$ less controlled. We illustrate this on the right-hand side of Fig. 8, where we showed the response of the electric field to a small external current. We see that even for $e^2 = 1$, deviation from linear response are seen for small external currents. Note also discretization artefacts are also expected to be stronger for larger charges [17].

The precision of our data allows us to look at the coupling- dependence of the resistivity. For simplicity, we fit only the data obtained from the quenches, as they are more numerous. We observe the dependence close to being quadratic but with clear subleading corrections. As shown on the figure, they are compatible with logarithmic corrections (our range of data is not large enough to distinguish it from a fractional power). This behavior, already reported in [18] for $\Gamma_A$ is interesting, as it is of the same functional form as the known subleading correction to $\frac{1}{\sigma}$.

# 5 Conclusion

In this work we performed a study of classical lattice simulations of electrodynamics coupled to charged matter (and – in the last section – an effective axial dynamics). We have shown that the dynamics of the plasma are in agreement with a recent formulation of MHD organized around the 1-form symmetry associated with the magnetic flux conservation.

A key point here is that the *resistivity* of the plasma – as determined from Kubo formulas arising from 1-form symmetry – remains finite and correctly predicts dynamical quantities such as the rate of magnetic field diffusion and axial charge relaxation.

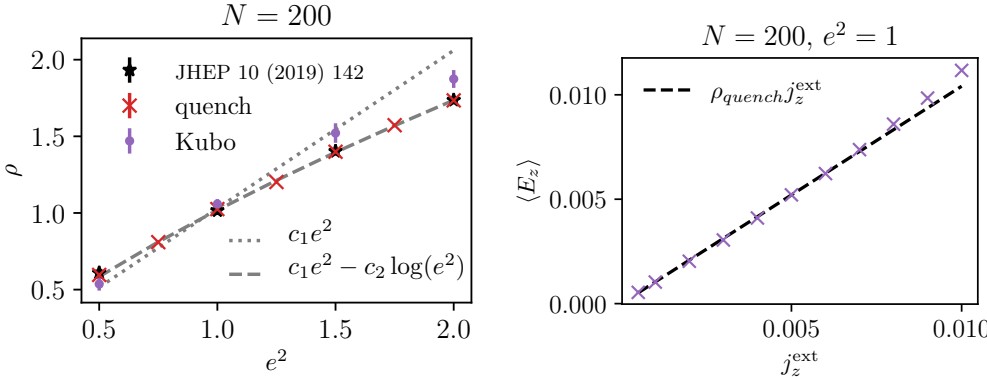

Figure 8: **Left:** Charge dependence of the resistivity for different methods. Black dots corresponds to extracting the resistivity from the chiral decay rates of [18]. The red dots are obtained by applying a constant external current to the system and fit the linear response of the electric field. The purple dots are obtained from the Kubo formula. The chiral decay rate data are indistinguishable from the linear response extraction. This is not surprising as both methods have similar systematics. The Kubo formula results are in good agreement a small charges but display a few percent tension at larger charges. We attribute this small discrepancy to small uncontrolled systematics (small region of validity of linear response and potential remaining finite volume effects). **Right:** Example of the extraction of the resistivity from the linear response to an external current. We perform simulations for different external currents and extract the linear response of the electric through a polynomial fit. We also note that the regime of validity of linear response seems relatively limited in this system.

A conventional formulation of the plasma would normally use the *conductivity* instead; here some care must be taken, as in a theory of dynamical electromagnetism, if the plasma is fully thermalized then the conductivity cannot be non-perturbatively defined in terms of its usual Kubo formula, as electrodynamic fluctuations drive the low-frequency limit of this formula to zero.

However if one can arrange a hierarchy of scales so that the electric field relaxation rate $\tau_E^{-1}$ is much slower than any other time-scale in the problem, then the appropriate correlation function for the electric current saturates at a constant value over an intermediate range of frequencies $\tau_E^{-1} \ll \omega \ll \Lambda$ and can be used to define the conductivity, which is then numerically equal to the inverse resistivity. Such a hierarchy is not present in the lattice simulations presented in this work. This hierarchy is however in principle present in weakly-coupled electromagnetism, and existing perturbative calculations of the conductivity in a theory of dynamical electromagnetism should presumably be interpreted in this context.

Indeed, if the electromagnetic coupling is small enough, we expect it to have little effect in the intermediate range of frequencies above, and the conductivity of the ungauged theory would then be essentially equal to the inverse resistivity in the gauged theory, as schematically illustrated in Figure 9, and implicitly assumed in much of the literature. We believe a completely convincing argument to this effect would require explicitly incorporating dynamical long-range electromagnetic fields in a microscopic transport calculation to show from first principles the crossover exhibited on hydrodynamic grounds in Eq (35).

Such considerations can have consequences on predictions for anomalous transport in systems with dynamical electromagnetic fields. We discussed the example of the chiral decay rate. Due to the presence of the chiral magnetic current induced by the anomaly, the chiral

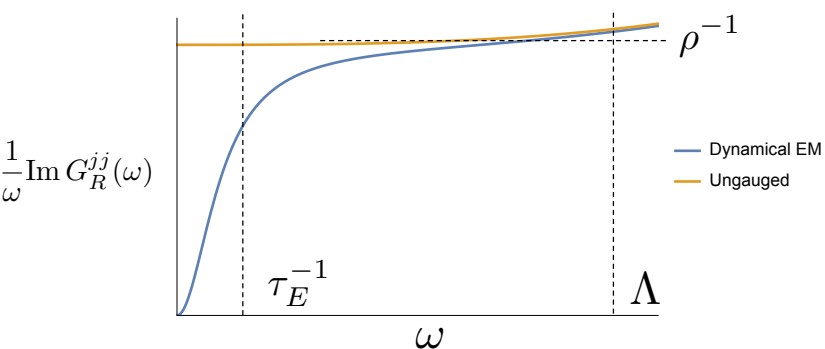

Figure 9: Schematic illustration of effect on current-current correlation function of weakly coupling dynamical electromagnetism to $U(1)$ global symmetry current. As shown in Section 3.2, correlator in theory with dynamical EM vanishes at low frequencies $\omega \ll \tau_E^{-1}$. If $e$ is sufficiently small, the effect at intermediate frequencies is expected to be small, and then the conductivity of the ungauged theory will determine the inverse resistivity of the gauged theory.

charge exponentially decays into gauge fields. We showed in this work that the rate previously measured in [17, 18] is completely consistent with hydrodynamic expectations, provided the transport coefficients are obtained from the Kubo formula derived in [15], thus resolving a previous confusion.

One might wonder whether we could also make a similar statements for axial charge relaxation in *non*-Abelian gauge theories. Here it is helpful to note that the key expression (40) is in fact a special case of the more general expression

$$\Gamma_A = \lim_{\Omega \to 0} \frac{k^2}{\chi_A \Omega} \mathrm{Im}\, G_{QQ}^R(\Omega, \vec{p} = 0), \tag{43}$$

where $G_{QQ}^R(\Omega, \vec{p})$ is the retarded correlation function of the topological density $Q(x)$, which is $Q = F_{\mu\nu}\tilde{F}^{\mu\nu}$ in the Abelian case (see [50] for more details[9]) and $\mathrm{Tr}(F_{\mu\nu}^a \tilde{F}^{a\mu\nu})$ in the non-Abelian case. In this work we exploited the continuous 1-form $U(1)$ symmetry in Abelian gauge theory to relate the above observable to a transport coefficient $\rho$. However in a non-Abelian plasma there is generally at most only a discrete $\mathbb{Z}_N$ 1-form symmetry, with no corresponding transport coefficient or universal hydrodynamic description, and we do not expect to be able to make any universal statements.[10]

One interesting direction for future research is the study of fluctuation effects in MHD. It is well-known that generically in hydrodynamics fluctuations can result in long-time tails in hydrodynamic observables [55, 56]. These are suppressed by loop factors and are not visible in classical calculations, and can result in non-analyticities in the $\omega$-dependence of various correlation functions. Numerically we do not currently see any smoking-gun evidence for such non-analyticity: e.g. the $\omega$-dependence of the gauged conductivity in Figure 2 appears to be well approximated by the classical $\omega^2$ dependence, though a small shift in the exponent would likely not be visible numerically. Theoretically we are not aware of much study of long-time tails in the relativtistic MHD context. One recent result is [50] in the context of chiral MHD, essentially showing that hydrodynamic loop corrections to the decay of the axial charge at zero

---

[9] [50] was released by a subset of the current authors after the first version of this work was placed on the arXiv.

[10] In the non-Abelian context the quantity computed in (43) is generally called the Chern-Simons diffusion rate, and represents the topologically induced axial charge dissipation stemming from the axial anomaly in non-Abelian theories. This quantity has been extensively studied both at weak-coupling [51–53] and from holography [54].

magnetic field result in non-analytic behavior in $\omega$ that is nevertheless irrelevant, consistent with the numerical results exhibited here. It would be very interesting to systematically study long-time tails in MHD using the techniques of [55,56] and confront the results with more detailed numerics at low frequencies.

Finally, the fact that the conductivity is a useful dynamical quantity only for weakly coupled matter is not well appreciated in the literature. As we have shown, this can have quantitative consequences. Our results suggest that it may make sense to reevaluate current descriptions of chirally assisted phenomena in cosmology, see for instance [40, 41, 57, 58] and references therein for a few examples. Similarly, recent developments considered the interplay of the chiral dynamics discussed in this work and on non-Abelian topology changing processes (sphalerons) [59]. The precise value of the chiral decay rate also impacts any resulting predictions.

## Acknowledgments

We would like to thank E. Grossi, D. Teaney, S. Grozdanov, S. Hartnoll and P. Kovtun for valuable discussions.

**Funding information** This research was supported in part by the National Science Foundation under Grant No. NSF PHY-1748958. This work was supported by the U.S. Department of Energy, Office of Science, Office of Nuclear Physics, Grant No. DE-SC0012704 (AF). NI is supported in part by STFC grant number ST/T000708/1, and is grateful to hospitality from the Kavli Institute for Theoretical Physics and the Institute for Nuclear Theory in UW during the course of this work. The work of NP is supported by the grant for development of new faculty (Ratchadapiseksomphot fund) and Sci-Super IX_66_004 from Chulalongkorn University. NP would also like to thank Leiden University, Durham University and NORDITA for their hospitality during the course of this project.

## A Correlators and the classical limit

We use this appendix to collect our conventions regarding correlators and make elaborate how classical correlators are related to the quantum ones. We will mostly follow the discussion presented in [60]. Out of equilibrium, two independent unequal time two-point functions can be defined for each operator $O$. They can be chosen as the statistical correlator $G_s^O(x; y)$ and the spectral correlator $G_\rho^O(x; y)$

$$G_s^{OO}(x; y) = \frac{1}{2}\{O(x)O(y)\}, \tag{A.1}$$

$$G_\rho^{OO}(x; y) = -i[O(x), O(y)]. \tag{A.2}$$

At the level of two-point functions, thermal equilibrium is expressed by the KMS relation, which relates the two correlators

$$G_s^{OO}(k) = \left(\frac{1}{e^{\omega/T} - 1} + \frac{1}{2}\right)\rho(k), \tag{A.3}$$

with $k = (\omega, \vec{k})$ and

$$\rho(k) = iG_\rho^{OO}(k), \tag{A.4}$$

the spectral function of the operator $O$, which is real and positive definite.

Transport properties are usually expressed in terms of the retarded correlator $G_R^{OO}$

$$G_R^{OO}(x;y) = \theta(x_0 - y_0)G_\rho^{OO}(x;y). \tag{A.5}$$

More precisely, transport can be read from the imaginary part of $G_R^{OO}$, which is nothing less than the spectral function itself

$$\mathrm{Im}(G_R^{OO}(k)) = -\frac{\rho(k)}{2}. \tag{A.6}$$

In the classical limit, the classical correlator is a good approximation to the statistical one $G_{cl}^{OO}(x) \approx G_s^{OO}$. It also captures transport properties, as the classical spectral function can be recovered through the KMS relation (A.3) in the classical limit

$$G_{cl}^{OO}(x) \approx \frac{T}{\omega}\rho(k). \tag{A.7}$$

Plugging this into (4)–(12), one gets the classical Kubo formula (21)-(22) (note that a factor of two is absorbed in the symmetric integration from zero to infinity).

# B   A global symmetry interpretation of weak coupling

Given a system with a conserved magnetic flux, when is it useful to think of it as Maxwell electrodynamics weakly coupled to charged degrees of freedom?[11]

In this Appendix we seek to give a universal definition of the concept of "weak electromagnetic coupling", relating it to emergent hydrodynamic timescales discussed in the main draft. We take a somewhat leisurely exposition here, taking opportunities to connect to elementary electrodynamics.

To orient ourselves, let us consider pure Lorentz-invariant electrodynamics, i.e. the action

$$S = \int d^4x \left( -\frac{1}{4e^2}F_{\mu\nu}F^{\mu\nu} + b_{\mu\nu}\epsilon^{\mu\nu\rho\sigma}F_{\rho\sigma} \right). \tag{B.1}$$

Here $b_{\mu\nu}$ is the coupling to the external source for the 2-form magnetic flux current $J^{\mu\nu} = \frac{1}{2}\epsilon^{\mu\nu\rho\sigma}F_{\rho\sigma}$. To interpet this source, write $F_{\mu\nu} = \partial_\mu A_\nu - \partial_\nu A_\mu$ and note that the coupling now takes the form $\int d^4x j_{\mathrm{ext}}^\mu A_\mu$, where as in (11) we have

$$j_{\mathrm{ext}}^\mu \equiv \epsilon^{\mu\nu\rho\sigma}\partial_\nu b_{\rho\sigma}. \tag{B.2}$$

In other words, the natural source for the 2-form current can be understood as a *fixed* external electric charge current. In conventional electrodynamics, this external source is often called the *free* charge current. In textbooks we often consider the 3-vector fields **H** and **D**, which the reader might (grudgingly) recall are *defined* as the objects who obey the bare Maxwell's equations sourced by the free charge current, i.e.

$$\nabla \times \mathbf{H} = \mathbf{j}_{\mathrm{ext}} + \frac{\partial \mathbf{D}}{\partial t}, \qquad \nabla \cdot \mathbf{D} = \rho_{ext}. \tag{B.3}$$

Comparing (B.3) with (B.2) we see that the components of the 2-form source $b_{\mu\nu}$ are actually precisely **H** and **D**:

$$b_{ti} = H_i, \qquad b_{ij} = \frac{1}{2}\epsilon_{ijk}D^k. \tag{B.4}$$

---

[11]We are grateful to S. Hartnoll for discussions related to the content of this section.

This identification involves derivatives and so is actually ambiguous up to the following transformation parametrized by an arbitrary 1-form:

$$b_{\mu\nu} \to b_{\mu\nu} + \partial_\mu \Lambda_\nu - \partial_\nu \Lambda_\mu, \tag{B.5}$$

which leaves invariant $j_{\text{ext}}$. It will nevertheless turn out to be helpful in relating the universal higher-form language with elementary concepts in textbook electrodynamics.

Now let us examine a slightly more general situation: consider a Coulomb phase of electrodynamics in a medium that is no longer Lorentz-invariant, and is expected to be characterized by two parameters $\epsilon$ and $\Xi$ (i.e. the electric and magnetic permeabilities). The Maxwell action is then modified to be:

$$S = \int d^4 x \left( -\frac{1}{4\Xi} F_{ij} F^{ij} - \frac{1}{2} \epsilon F_{ti} F^{ti} + \frac{1}{2} b_{\mu\nu} \epsilon^{\mu\nu\rho\sigma} F_{\rho\sigma} \right). \tag{B.6}$$

It is instructive to write the general expression for the 2-form current $J^{\mu\nu}$ in the presence of $b$. After a short computation we find

$$J^{ti} = \Xi \left( b^{ti} + \partial^t \tilde{A}^i - \partial^i \tilde{A}^t \right), \qquad J^{ij} = \frac{1}{\epsilon} \left( b^{ij} + \partial^i \tilde{A}^j - \partial^j \tilde{A}^i \right). \tag{B.7}$$

We obtained this expression by varying the action with respect to the ordinary photon $A$ and then parametrized the solution to the resulting equation of motion in terms of a *new* dynamical vector field $\tilde{A}$ – one can think of this as the dual magnetic photon.

The form of the currents that one obtains from here should be familiar: it is precisely the 1-form version of a spontaneously broken symmetry, with $\tilde{A}$ being the 1-form Goldstone mode. Indeed it is well-known that the ordinary Coulomb phase of electrodynamics is the phase where the 1-form magnetic flux symmetry is spontaneously broken [1, 21, 22]. Comparing this to an ordinary (0-form) spontaneously broken symmetry, we see that $\Xi_b$ is the 1-form charge susceptibility, whereas $\frac{1}{\epsilon}$ plays the role of the 1-form superfluid stiff-ness.

Let us now connect to elementary electrodynamics. Using (B.3) and expressing the two-form current $J^{\mu\nu}$ in terms of the regular **E** and **B**[12] fields we find:

$$\mathbf{B} = \Xi \mathbf{H}, \qquad \mathbf{E} = \frac{1}{\epsilon} \mathbf{D}, \tag{B.8}$$

which are the usual relations relating the magnetic and electric fields to **D** and **H**. The main point to note here is that the macroscopic electric and magnetic permeability $\epsilon$ and $\Xi$ have a precise meaning in terms of the thermodynamic parameters characterizing the spontaneous breaking of 1-form symmetry in a material. Indeed the speed of the gapless photon is

$$c^2 = \frac{1}{\epsilon \Xi}, \tag{B.9}$$

which can now be re-interpeted as the usual expression for the speed of the Goldstone mode in a conventional symmetry broken phase with stiffness $\epsilon^{-1}$ and susceptibility $\Xi$ (see e.g. [61]).

Now, let us consider what it means to add dynamical charges to this system. We can consider modifying the action as follows:

$$S = \int d^4 x \left( -\frac{1}{4\Xi} F_{ij} F^{ij} - \frac{1}{2} \epsilon F_{ti} F^{ti} + \frac{1}{2} b_{\mu\nu} \epsilon^{\mu\nu\rho\sigma} F_{\rho\sigma} + j_\mu^{\text{dyn}} A^\mu \right), \tag{B.10}$$

where here $j_\mu^{\text{dyn}}$ is an extra density of *dynamical* charges. In their absence the system is in a Coulomb phase; once they are present the system may be in a different phase. Thus in a

---

[12]Note that the shift of $b_{\mu\nu}$ by $\partial_{[\mu} \tilde{A}_{\nu]}$ is exactly the ambiguity (B.5).

universal sense one can simply imagine that adding charges has the effect of disordering the spontaneous breaking of 1-form symmetry in the Coulomb phase.

To proceed we need a choice for their dynamics. We now specialize to the choice that is relevant for plasma, i.e. we assume that there exists a parameter $\sigma$ such that the following relation is true:

$$j_{\text{dyn}}^i = \frac{1}{2}\sigma\epsilon^{ijk}J_{jk}. \tag{B.11}$$

This is stating that if the gauge potential $A$ is frozen, then the electrical charge current $j_{\text{dyn}}^i$ responds to the application of an electric field with 0-form conductivity $\sigma$. The equation of motion that one finds from here is now simply:

$$\frac{1}{\Xi}\epsilon^{ijk}\partial_j J_{tk} + \epsilon\partial_t\epsilon^{ijk}J_{jk} = j_{\text{dyn}}^i. \tag{B.12}$$

This is simply the in-medium Maxwell equation, which we have written in terms of $J^{\mu\nu}$ so that it may be related to (31), which we recall made no mention of any microscopic description. We find the following matching of parameters:

$$\tau_E = \frac{\epsilon}{\sigma}, \qquad \rho = \frac{1}{\sigma}. \tag{B.13}$$

In other words, by viewing the plasma as a deformation of a phase where the 1-form symmetry is spontaneously broken (i.e. the free photon phase), we have obtained information about an extra non-universal scale $\tau_E$, expressed in terms of thermodynamic data $\epsilon$ characterizing the spontaneously broken phase. At times $t \gg \tau_E$ we obtain a description in terms of MHD alone.

Finally, let us note that for a system where the electrodynamic sector alone (i.e. in the absence of $j_{\text{dyn}}$) is Lorentz-invariant, then from (B.9) we find $\epsilon = \Xi^{-1}$, and the electrodynamic number is characterized by a single number, the electromagnetic coupling $e$; further comparing (B.6) to (B.1) we see that $\Xi = e^2$, and we find

$$\tau_E = \frac{1}{\sigma e^2}, \qquad \rho = \frac{1}{\sigma}. \tag{B.14}$$

Note in particular that as the electromagnetic coupling $e$ is taken to zero, $\tau_E$ grows and the MHD description's range of validity $t \gg \tau_E$ is smaller, as one might expect.

## C Numerics

We summarize briefly here our numerics, more information can be found in [17,18]. See for instance [29] for a review on classical lattice techniques. The thermal initial conditions for the system are sampled thanks to a standard Metropolis algorithm applied to (15). At this stage, Gauss law is only mildly satisfied. We remedy this situation by *cooling* the system. The configuration generated by the Monte-Carlo is brought to the closest configuration satisfying Gauss law through steepest descent. For simulations with a background magnetic field, we impose non-zero fluxes by using the twisted boundary conditions described in [18]. The external electric current and electric fields scenario described in the main text are realized as quenches; they are turned on at the beginning of the time evolution. The same is true for the chiral chemical potential.

Concretely, we solve the following discretized set of equations

$$\partial_t \varphi = \pi, \qquad \partial_t \pi = \sum_i D_i^- D_i^+ \varphi - V_{,\varphi^*}, \tag{C.1}$$

$$\partial_t A_i = E_i + E_i^{ext}, \qquad \partial_t E_i = 2e^2 \mathrm{Im}\{\varphi^* D_i^+ \varphi\} - \sum_{j,k} \epsilon_{ijk} \Delta_j^- B_k - \frac{1}{2\pi^2} \mu_A B_i^{(8)} + j_i^{ext}, \tag{C.2}$$

$$\mu = \partial_t a, \qquad \partial_t \mu = \frac{3}{2\pi^2} \frac{1}{T^2} \frac{1}{N^3} \sum_{\vec{n}} \frac{1}{2} \sum_i E_i^{(2)} \left( B_i^{(4)} + B_{i,+0}^{(4)} \right), \tag{C.3}$$

with $\Delta_\mu^\pm f = \pm \frac{1}{\mathrm{d}x}(f_{\pm\mu} - f)$, $D_\mu^\pm f = \pm \frac{1}{\mathrm{d}x}(e^{\mp ie\mathrm{d}x^\mu A_\mu(n\pm\frac{1}{2})} f_{\pm\mu} - f)$ the forward/backward finite difference operator and covariant derivatives, $B_i = \sum_{jk} \epsilon_{ijk} \Delta_j^+ A_k$, and

$$E_i^{(2)} \equiv \frac{1}{2}(E_i + E_{i,-i}), \tag{C.4}$$

$$B_i^{(4)} \equiv \frac{1}{4}(B_i + B_{i,-j} + B_{i,-k} + B_{i,-j-k}), \tag{C.5}$$

$$B_i^{(8)} \equiv \frac{1}{2} \left( B_i^{(4)} + B_{i,+i}^{(4)} \right), \tag{C.6}$$

as composite operators necessary to have a proper discretization $\vec{E} \cdot \vec{B}$ as a total derivative.

The external sources $E_i^{ext}$ and $j_i^{ext}$ are used for our linear response analysis. As already mentioned in the main text, adding an external current $j_i^{ext}$ is unambiguous. On the other hand, the meaning of an external electric field $E_i^{ext}$ in the dynamical system is less clear. We implement it as a shift in the momentum operator. Its effect is to effectively change the initial conditions for the gauge field and force the system out of thermal equilibrium. It is worth nothing that when $E_i^{ext}$ is applied the 1-form symmetry current $J^{ij}$ is proportional to $E - E^{ext}$.

To perform the time evolution, we use a simple leapfrog scheme, detailed in [17].

# D   More on classical field theory in thermal equilibrium

We use this appendix to clarify the meaning of a classical thermal equilibrium for the reader not used to thinking about this problem.

A standard classical field theory cannot be in thermal equilibrium in the continuum, this is the standard Rayleigh-Jeans UV-catastrophe of classical field theory. The full quantum theory is regulated by generating the Bose-Einstein/Fermi-Dirac distribution instead of the Boltzmann. Technically, this can be seen in Euclidean time as the periodicity in time.

While this is the way nature appears to regulate thermal states, this is not unique. A classical field theory on a lattice with lattice spacing $a$ is also UV finite, even though the UV cutoff $a$ cannot be removed in a meaningful way.

Table 1: Lattice parameters.

| $e^2$ | # confs. Kubo |
|-----|---------------|
| 0.5 | 50 |
| 1 | 500 |
| 1.5 | 250 |
| 2 | 249 |

To illustrate these ideas, let us compute the thermal mass of a massless scalar field in a quartic potential. From Chapter 3 of [62], we have the thermal correlator at finite temperature for the $\lambda\phi^4$ theory, in momentum space, as:

$$\Delta(i\omega_n, k) = \Delta^{(F)}(i\omega_n, k)\left[1 - \left(\frac{\lambda T}{2}\sum_n \int \frac{d^3 k'}{(2\pi)^3}\Delta^{(F)}(i\omega_n, k')\right)\Delta^{(F)}(i\omega_n, k)\right], \qquad (D.1)$$

where $\omega_n$ are Matsubara frequencies and $k$ denotes the spatial 3-momentum. The above can be readily derived from a Feynman diagram of the following form,

$$\underline{\quad\quad} = \underline{\quad\quad} + \underset{\bullet}{\overset{\bigcirc}{\underline{\quad\quad}}} + \mathcal{O}(\lambda^2), \qquad (D.2)$$

where the term on the left hand side is the full propagator, the first term in the right hand side is the free propagator and the second term is the $\mathcal{O}(\lambda)$ correction to it.

The correction to the self-energy is $\Pi(i\omega_n, k)$ using, $\Delta^{-1}(i\omega_n, k) = \left(\Delta^{(F)}\right)^{-1} + \Pi(i\omega_n, k)$. From (D.1) we obtain,

$$\Delta(i\omega_n, k)^{-1} = \left(\Delta^{(F)}(i\omega_n, k)\right)^{-1}\left[1 + \left(\frac{\lambda T}{2}\sum_n \int \frac{d^3 k'}{(2\pi)^3}\Delta^{(F)}(i\omega_n, k')\right)\Delta^{(F)}(i\omega_n, k) + \mathcal{O}(\lambda^2)\right], \quad (D.3)$$

which in position space becomes,[13]

$$\Delta^{-1} = \left(\Delta^{(F)}\right)^{-1} + \frac{\lambda T}{2}\Delta^{(F)}(z = 0), \qquad (D.4)$$

where $z = 0$ is the point where the loop intersects the line, the black dot, as shown in the second term on the RHS of (D.2).

Now we put the above theory on a lattice with lattice spacing 'a' and compute the self-energy from above as, upto $\mathcal{O}(\lambda^2)$,

$$\Pi = \frac{\lambda T}{2}\Delta^{(F)}(z = 0) = \frac{\lambda T}{4\pi^2}\int_0^{k_{\max}} dk\, \frac{k^2}{k^2 + m^2} \overset{m \to 0}{=} \frac{\lambda T}{4\pi^2}k_{\max} = \frac{\lambda T}{4\pi^2}\frac{2\pi}{a} = \frac{\lambda T}{2\pi a}. \qquad (D.5)$$

We see that now the UV cutoff, which should be in this case interpreted as some scale in the problem, enters the results. Instead of the expected "quantum" $T^2$ dependence, we obtain $T/a$.

The plasmon peak we observe in the right side of Fig. 2 is conceptually of the same character; the lattice cutoff induces "classical" hard thermal loop. They can in principle be computed analytically. We abstained as it did not have direct relevance to our results.

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
