# Peer review of "Higher-form symmetry and chiral transport in real-time Abelian lattice gauge theory"

_SciPost Physics, doi:SciPost Phys. 17, 085 (2024)_

## Round 1 · Referee Report · Anonymous (Referee 1) · 2024-8-28

Strengths

1- Explicit and independent checks regarding the lack of meaning of conductivity in strong coupling regimes. 2- Application of the developed formalism to chiral transport. 3- Clarity in the explanations.

Report

The authors employ the magnetohydrodynamics (MHD) formalism to study the effective evolution of electrodynamics coupled to charged matter at finite temperature, further employing lattice techniques to assess the microscopic properties of the system and compute the transport coefficients. Very interestingly, it is argued and checked that the notion of electric conductivity loses its meaning in the strong coupling regime, or more generically when a certain separation of scales is not present in the system. Furthermore, they apply their formalism to a chiral system and compute the relaxation rate for the axial charge arising from the ABJ anomaly.
Finally, the authors have successfully addressed the previous concerns in the report.
Overall, the paper provides interesting and innovative insight and deserves publication.

Recommendation

Publish (easily meets expectations and criteria for this Journal; among top 50%)

---

## Round 1 · Author Response

We would like to thank both the referees for the positive feedback, the careful reading and the insightful questions and comments; we have addressed them below and we think that they have improved the paper.

Below, in the list of changes, we address the questions raised by both referees and the change made in the manuscript.
We hope that the manuscript now satisfies the publication standard of SciPost.

---

## Round 1 · List of Changes

Warnings issued while processing user-supplied markup:

  • Inconsistency: plain/Markdown and reStructuredText syntaxes are mixed. Markdown will be used.
    Add "#coerce:reST" or "#coerce:plain" as the first line of your text to force reStructuredText or no markup.
    You may also contact the helpdesk if the formatting is incorrect and you are unable to edit your text.

From Referee 1

  1. In page 8 it is stated that regulating the classical field theory in thermal equilibrium via discretising space or via QFT are "obviously inequivalent" while "They do however belong to the same universality class". Both statements are in tension since it is not clear what is meant in this context by belonging to the same universality class. Indeed the authors themselves point out a non-trivial difference between both approaches in Appendix D by computing the thermal mass of the scalar field and obtain different scalings with temperature. In addition, they also discuss that the separation of scales required to have a well defined conductivity is "not present in the lattice simulations presented in this work" (page 25), while it should be present in the weak coupling QFT approach. I believe that the authors should review and explicitly state the limitations in comparing both theories.

Our response: We agree with the referee the language we used was not particularly clear. What we mean by universal is that both theories are described in the IR by the same EFT, namely MHD. We clarified this point as follow. We removed the sentence on page 8 " They do however belong to the same dynamical universality class. " and replaced it by "These theories are different in the UV but have the same global symmetries and are described by the same effective theory of MHD at long distances, and are thus in the same dynamic universality class. In particular, this means that the long distance physics of both theories will be the same at the qualitative level. For instance, they both exhibit the same kind of transport phenomena, magnetic flux diffusion, etc. which are described by the same Kubo formulas. The differences between the two theories manifest themselves as different matching coefficients, i.e. {\it a priori} different numerical values for the transport coefficients, which are determined in a complicated manner by the UV definition of the theory and the couplings in the potential. "

2- The authors compare their results to the conductivity computed in [48] and report an O(10) discrepancy. However, in [48] employs QFT techniques as opposed to the lattice regularisation. Once again, both approaches do not seem to be directly comparable, as the lattice regularisation lacks of the separation of scales required to defined the conductivity as explained in the submitted paper. This issue should be clarified in accordance with the first proposed change above.

Our response: We also agree with the referee that a detailed comparison here is not really possible due to the different UV completion. We have removed the emphasis on the O(10) discrepancy and merged some of this content into the subsequent paragraph on p27-28, which now reads:

"Finally, the fact that the conductivity is a useful dynamical quantity only for weakly coupled matter is not well appreciated in the literature. As we have shown, this can have quantitative consequences. Our results suggest that it may make sense to reevaluate current descriptions of chirally assisted phenomena in cosmology, see for instance [40, 41, 57, 58] and references therein for a few examples. Similarly, recent developments considered the interplay of the chiral dynamics discussed in this work and on non-Abelian topology changing processes (sphalerons) [59]. The precise value of the chiral decay rate also impacts any resulting predictions.

3- In page 8, it is stated that "the universal dynamics that we will discuss does not depend on the precise form of the potential". I would suggest that the authors are more explicit with what the mean by "universal dynamics" in this context and provide some form of justification as to why the form of the potential is apparently not relevant.

Our response: We believe this is now answered in the detailed explanation of the meaning of the long-distance physics given in the answer to #1: as long as we stay out of the Higgs phase, the couplings in the potential will determine the values of transport coefficients but will not change the fact that long-distance physics is given by MHD.

  1. In page 4 it is stated that the chiral "decay rate is controlled by the resistivity". Indeed this is shown in the paper in the context of abelian gauge theories. It is a genuine question whether a similar statement could be made for the topologically induced axial charge dissipation stemming from the axial anomaly in non-abelian theories.

Our response: We thank the referee for this question and have now addressed it in the conclusion on p27 -- in the paragraph containing Eq.(5.1). It reads:

" One might wonder whether we could also make a similar statements for axial charge relaxation in {\it non}-Abelian gauge theories. Here it is helpful to note that the key expression Eq.(4.4) is in fact a special case of the more general expression

$$ \Gamma_{A} = \lim_{\omega \to 0} \frac{k^2}{\chi_{A} \omega} \text{Im }G^{R}_{QQ} (\omega,\vec{p} = 0) $$
where $G^{R}_{QQ}(\omega,\vec{p})$ is the retarded correlation function of the topological density $Q(x)$, which is $Q = F_{\mu\nu} \tilde{F}^{\mu\nu}$ in the Abelian case (see [50] for more details) and $\text{Tr}(F^{a}_{\mu\nu} \tilde{F}^{a\mu\nu})$ in the non-Abelian case. In this work we exploited the continuous 1-form $U(1)$ symmetry in Abelian gauge theory to relate the above observable to a transport coefficient $\rho$. However in a non-Abelian plasma there is generally at most only a discrete $\mathbb{Z}_N$ 1-form symmetry, with no corresponding transport coefficient or universal hydrodynamic description, and we do not expect to be able to make any universal statements. In the non-Abelian context the quantity computed in Eq.(5.1) is generally called the Chern-Simons diffusion rate, and represents the topologically induced axial charge dissipation stemming from the axial anomaly in non-Abelian theories. This quantity has been extensively studied both at weak-coupling [51]-[54]. "

  1. A major difficulty in interpreting the results of the paper comes from the fact that dimensionful quantities are presented without units. This happens in the figures and more importantly in the estimation of resistivity in page 12 and the computation of the chiral decay rate in page 22 (fig. 6). This complicates the direct comparison with the results of [48] as well as with the chiral decay rate as computed for non-abelian theories existing in the literature. I kindly ask the authors to remedy this situation.

Our response: We apologize for this. Everything is expressed in units of the temperature. It appears we omitted this point in the draft. We also note that the dependence on the UV cutoff cannot be removed and thus a comparison of non-universal data to continuum QFT is not possible. We now explain this in detail in some paragraphs on p10, which read:

" In practice, and for the rest of this work, we work in units where the lattice spacing $a = 1$, and we set the temperature to $T = 1/a = 1$, i.e. our temperature is at the lattice scale. Let us briefly explain how to restore units to the dimensionless numerical results presented. Consider an observable $\mathcal{O}$ with mass dimension $\Delta$. On general grounds its functional dependence on all parameters will be given by

$$ \mathcal{O} = T^{\Delta} f(Ta, ma, \cdots) $$
where $f$ is a dimensionless function of all physical quantities measured in units of the lattice scale. Our results should be interpreted as determining the dimensionless function $f$ at a particular value of its arguments; the appropriate power of $T$ can be restored by dimensional analysis if required -- e.g. if we restore units to the bottom panel of Figure 1 it would be interpreted as a plot of $\rho T$ against $t T$ -- but as we are not in a continuum limit we stress that the dependence on the UV cutoff $a$ appearing in the scaling function $f$ can never be removed.

We also wish to fix the scalar mass $m$ and the coupling $\lambda$. For our problem, the only important consideration is that the parameters land the model in its unbroken phase. In order to ease comparison with Ref. [18], we adopt the same choice, namely $m^2 = e^2T^2/4$ and $\lambda=\frac{e^2}{2}$. This choice is motivated by phenomenological considerations and further discussed in Ref.~[18] "

6- The boundary conditions used in the lattice are relevant for the existence of absence of non-zero fluxes. In page 30 the authors imply that they use twisted boundary conditions in certain cases while in page 10 (bottom of Sec 2) they suggest the use of periodic boundary conditions. It would be useful to have a clear statement as to when the authors choose either of the boundary conditions.

Our response: We only use twisted boundary conditions when there is a background magnetic field, i.e. only in Section IV B. We now clarify this by adding a paragraph on p9 which reads:

" We use periodic boundary conditions for the fields φn, πn and En. In the absence of external magnetic field, An also has periodic boundary conditions. In the cases where we consider a background magnetic field – which is only in Section IV B – we implement it through twisted boundary conditions for the An fields. We refer the interested reader to Appendix A of [29] for technical details."

7- Typos: In page 11, Sec. III A, there is a reference to the upper left hand side of Fig.5 which seems to be wrong (there is a single figure in Fig 5, no notion of upper lower left or right). The description does not match what is shown in Fig. 5 either. The same happens in page 12. I would kindly ask the authors to check that all the hyperlinks included refer to the proper object. Page 5 line 1 "an conventional" Page 23 "showed the the response"

The typos are now fixed and we thank the referee for pointing them out.

From Referee 2:

  1. I suggest that the authors address the following question. In usual fluctuating hydrodynamics, the fluctuations lead to the long-time hydrodynamic tail (see, e.g., https://arxiv.org/abs/hep-th/0303010v1). What is the analog of the hydrodynamic tail in MHD (Chiral MHD)? Can one observe it from numerical simulations?

Our response: We thank the referee for the interesting question. We have addressed it in a new paragraph on p27 which reads:

" One interesting direction for future research is the study of fluctuation effects in MHD. It is well-known that generically in hydrodynamics fluctuations can result in long-time tails in hydrodynamic observables [55, 56]. These are suppressed by loop factors and are not visible in classical calculations, and can result in non-analyticities in the $\omega$-dependence of various correlation functions. Numerically we do not currently see any smoking-gun evidence for such non-analyticity: e.g. the $\omega$-dependence of the gauged conductivity in Figure 2 appears to be well approximated by the classical $\omega^2$ dependence, though a small shift in the exponent would likely not be visible numerically. Theoretically we are not aware of much study of long-time tails in the relativtistic MHD context. One recent result is [50] in the context of chiral MHD, essentially showing that hydrodynamic loop corrections to the decay of the axial charge at zero magnetic field result in non-analytic behavior in $\omega$ that is nevertheless irrelevant, consistent with the numerical results exhibited here. It would be very interesting to systematically study long-time tails in MHD using the techniques of [55, 56] and confront the results with more detailed numerics at low frequencies. "

---

## Editorial Decision

published